# Does the Association of Sedentary Time or Fruit/Vegetable Intake with Central Obesity Depend on Menopausal Status among Women?

**DOI:** 10.3390/ijerph191610083

**Published:** 2022-08-15

**Authors:** Jing Su, Qingting Li, Ping Mao, Hua Peng, Huiwu Han, James Wiley, Jia Guo, Jyu-Lin Chen

**Affiliations:** 1Department of Nursing, Medical College, Shantou University, Shantou 515041, China; 2Department of Nursing, School of Medicine, Zhejiang University City College, Hangzhou 310015, China; 3Xiangya School of Nursing, Central South University, Changsha 410017, China; 4Department of Nursing, The Third Xiangya Hospital of Central South University, Changsha 410013, China; 5Department of Nursing, Xiangya Hospital of Central South University, Changsha 410013, China; 6Institute for Health Policy Studies, University of California San Francisco (UCSF), San Francisco, CA 94158, USA; 7Nursing School, Xinjiang Medical University, Urumqi 830054, China; 8Department of Family Health Care Nursing, School of Nursing, University of California San Francisco (UCSF), San Francisco, CA 94143, USA

**Keywords:** central obesity, sedentary time, fruit and vegetable intake, menopausal status, women

## Abstract

(1) Background: The prevalence of central obesity is growing rapidly among women, and the prevalence differs by menopausal status. Longer sedentary time and less fruit/vegetable (F/V) intake increased the risk of central obesity. Among women of different menopausal statuses, controversy surrounds the association between sedentary time or F/V intake and central obesity. This study aimed to explore whether the independent and joint associations between sedentary time or F/V intake and menopausal status are correlated with central obesity, respectively. (2) Methods: This cross-sectional study was conducted in Hunan, China. Self-reported questionnaires were used to gather information on demographic characteristics, menopausal status, sedentary time, and F/V intake. Waist circumference was measured at the study site. Binary logistic regression and multiple interaction models were used to explore the independent and joint associations of menopausal status and the above two lifestyle variables with central obesity. (3) Results: A total of 387 women with a mean age of 47.7 ± 6.6 years old participated in the study. The prevalence of central obesity was 52.8%. Peri- and post-menopause statuses and not taking five servings of F/V per day were risk factors of central obesity (*p* < 0.05), whereas no significant association was found between sedentary time and central obesity (*p* > 0.05). Among peri-menopausal (mutual odds ratio (OR): 2.466, 95% confidence interval [CI]: 0.984–6.182; *p* < 0.05) and post-menopausal women (mutual OR: 2.274, 95% CI: 1.046–4.943; *p* < 0.05), more than 4 h of sedentary time per day was associated with a high risk of central obesity. Among pre-menopausal women, the consumption of five servings of F/V per day was associated with a low risk of central obesity (mutual OR: 0.444, 95%CI: 0.236–0.837, *p* < 0.05). (4) Conclusions: More than half of women in the central south of China presented with central obesity, spent >4 h of sedentary time per day, or did not engage in recommended F/V intake. Healthier lifestyle intervention is warranted to prevent central obesity development, including reducing the sedentary time to <4 h per day for peri- and post-menopausal women, while increasing taking five servings of F/V per day for pre-menopausal women.

## 1. Introduction

Central obesity, also known as abdominal obesity, refers to the accumulation of fat in the abdominal area, particularly due to excess visceral fat [1]. Central obesity is often determined by waist circumference (WC, cm), which is more closely associated with chronic diseases (such as hepatocellular carcinoma [2], diabetes [3], and vascular complications [4]) compared with generalized obesity, which is identified by body mass index (BMI, kg/m^2^). The prevalence of central obesity is higher among women than men and has steadily increased worldwide in recent decades [5]. In the U.S., central obesity among women has increased to 67.8% in 2013–2014 (annual increase rate: 0.75%), and 80.0% of women are projected to have central obesity by 2030 [6]. In China, the prevalence of central obesity was 51.7% in women, and it increased from 27.8% to 51.7% from 1993 to 2011 [5,7,8]. Women with central obesity are at a high risk of developing breast cancer (1.09-fold) [9], endometrial cancer (1.21–1.27-fold) [10], polycystic ovary syndrome (1.73-fold) [11], and psychological problems, such as psychological distress (1.31-fold) [12], anxiety (1.99-fold) [13], and depression (1.28-fold) [14], which impose a heavy disease burden on families and the society [15]. Therefore, central obesity must be prevented or reversed, especially among women. 

Before developing any targeted prevention or reversal intervention, a comprehensive understanding of central obesity-related risk factors in the target population is a prerequisite. Central obesity in women is caused by a variety of factors, including genetic (obesity-associated gene [16]), behavioral (dietary habits, alcohol consumption, physical activity, and smoking [17]), and socioeconomic factors (age, ethnicity, and socioeconomic level [6,18]). Compared with men, menopausal status is a unique physiological characteristic of women. Menopausal status refers to the different stages of reproductive aging in women [19]: pre-menopausal phase, also known as the growth period, starts from female menstrual cramps and progresses from irregular to a regular menstrual process, which is based on the bleeding pattern with regular menstrual cycles. Peri-menopause is a midlife transition state experienced by women and characterized by changes in the women’s baseline for at least two cycles to 11 months of amenorrhea [20]. Post-menopause refers to the period from the last menstruation to the end of life and in which women have amenorrhea for at least 12 months. Peri- and post-menopausal women have a high risk of developing central obesity [21]. Given the changes in reproductive hormones, women at the peri-menopausal status are prone to a rapid increase in fat mass and redistribution of fat to the abdomen [22,23]. The prevalence of central obesity is higher in post-menopausal women compared with pre-menopausal women, possibly due to the reduced energy expenditure. However, in a sample of 3064 women in the U.S., the study reported that menopausal status was not significantly associated with weight gain and WC due to the interference of chronological age. However, the result was limited to three years of follow-up [24]. Because of the inconsistent findings in current scientific literature, more studies are needed to explore the association between menopausal status and central obesity after controlling for confounding factors such as age while examining the possible behavior risk factors.

In addition to menopausal status, several behavioral factors, such as physical activity and diet behavior, are the major driving forces underlying central obesity [25,26]. Previous studies advocated an increase in physical activity to prevent central obesity. However, other research revealed that prolonged sedentary time, which is independent of walking time [27], increased the risk of central obesity in women, which is consistent with findings from the National Health and Nutrition Examination Survey of America [28]. Sedentary time refers to the time spent in any waking behavior characterized by an energy expenditure of 1.5 metabolism equivalents or lower while sitting, reclining, or lying [29]. Sedentary time may play different roles in the development of central obesity across different menopausal statuses [30,31,32]. For pre-menopausal women in Asia (mean age: 20.5 years), sedentary time has not been associated with central obesity [30]. Whether a prolonged sedentary time can increase the risk of central obesity remains controversial for post-menopausal women. A study from Spain showed that an extended sedentary time increased the risk of developing central obesity [31], whereas research from Korea reported a non-significant correlation between the two variables [32]. For peri-menopausal women, the data on correlations between sedentary time and central obesity are lacking. Thus, more studies are needed to describe the association between sedentary time and central obesity among women of different menopausal statuses. 

The World Health Organization recommends that adults consume five or more cups of F/V per day or 400 g/day to prevent obesity [33]. Previous studies showed that more than four servings of F/V intake per day were associated with a low risk of central obesity in adult women [34]. However, F/V intake may play different roles in the development of central obesity among women across different menopausal statuses [35,36]. For pre-menopausal women, a study from Korea indicated that a high F/V intake per day was associated with a low risk of central obesity [35]. Still, another Isfahan research did not find significant correlations among women aged < 30 years [36]. For post-menopausal women, no significant correlation was observed between F/V intake and central obesity [35]. Considering the controversial findings among pre-menopausal women, the inconsistent findings in post-menopausal women, and the absence of data among peri-menopausal women, more studies are needed to determine the association between F/V intake and central obesity in women at different menopausal statuses. 

In summary, menopausal status, sedentary time, and F/V intake play essential roles in central obesity development among women. Currently, the evidence suggests an inconsistent association between sedentary time or F/V intake and central obesity among women of pre- and post-menopausal statuses. A limited number of studies have reported associations among peri-menopausal women. To provide evidence for appropriate prevention strategies against central obesity in women of different menopause statuses, we aimed to (1) describe the current situation of central obesity, sedentary time, and F/V intake among women in the central south of China; (2) explore whether the independent(menopausal status, F/V intake and sedentary time) and joint associations(menopausal status × F/V intake and menopausal status × sedentary time) between sedentary time or F/V intake and menopausal status are correlated with central obesity, respectively. 

## 2. Materials and Methods

### 2.1. Study Design, Settings, Recruitment, and Participants

This research was a cross-sectional study conducted in Hunan province, which is located in the central-south region of China. This area is one of the major regions in China with diverse geographical styles and cultures It has a dense population and the highest economic development in China. Participants were recruited from three county-level general hospitals and two central township hospitals, where the universal cancer screening programs (breast and cervical cancer) for female residents funded by the Chinese government in Hunan province in China were implemented from 30 July to 26 October 2018. When the potential participants came to the research sites for a free screening, the trained research assistants explained the purposes and procedures of the study. Women were invited to participate with informed consent if they were eligible and agreed to participate. This study followed the guidelines of Strengthening the Reporting of Observational Studies in Epidemiology Checklist for cross-sectional studies (STROBE 2020), which aimed to ensure the quality and specification of study reporting [37]. This study was approved by the Human Research Committee at the Central South University in China (No. 2018028).

The inclusion criteria for the participants comprised the following: (1) of the female biological sex, (2) aged 35–64 years old for inclusion criteria of the universal breast and cervical cancer screening programs, (3) able to speak Mandarin, and (4) residents of the research sites. The exclusion criteria were as follows: (1) inability to perform regular daily activities due to serve disease, such as uncontrolled chronic heart-related conditions, any major signs of cardiovascular, pulmonary, or metabolic disease, had two or more major coronary risks factors, or cancer and renal failure, (2) inability to walk without a cane or other assistive device, (3) pregnancy; (4) being on a ketogenic diet due to diseases, on a strict diet, or having eating disorders such as anorexia, bulimia, etc. A power analysis of the sample size was conducted to ensure an adequate sample size for this study. The OR associated with the interaction is about 0.45 for central obesity or 2.22 for no central obesity. For a logistic regression with *n* = 387, a two-tailed null hypothesis test with an odds ratio (OR) of 1.0 was conducted for a given predictor against an alternative OR of 0.45 with a power of 0.99. Assuming that α = 0.05, the base prevalence of central obesity was 0.20, and the R2 with other predictors was 0.50 (G*Power 3.1, Heinrich Heine University Düsseldorf, Düsseldorf, Germany). 

### 2.2. Measurement

#### 2.2.1. Sociodemographic Characteristics 

Self-reported questionnaires designed by the research team were used to collect information on the participants’ age, marital status, education level, family income, occupation, medical history, smoking, and alcohol consumption. 

#### 2.2.2. Menopausal Status

According to the 2016 Stages of Reproductive Aging Workshop [19], menopausal status in this study was self-reported and obtained by interview by the research assistants. The stages of menopause used in this study were pre- (no change in frequency of menstrual periods), peri- (increased irregularity and decreased predictability of menstrual periods), and post-menopause (complete cessation of menstrual periods) [38].

#### 2.2.3. Sedentary Time 

The self-reported survey was used to ask participants whether they engaged in >4 h of sedentary time. The sedentary time was estimated by asking them the question, “How many hours do you sit per day in the past two months (including time spent at work or home, sedentary, lying down at a desk, watching TV, watching the mobile phone, playing mahjong, etc.)?” which was adapted from the International Physical Activity Questionnaire (Spearman’s rho clustered around 0.8) [39]. Daily sedentary time > 4 h was defined as an unhealthy sedentary behaviour [27] in this study. 

#### 2.2.4. F/V Intake

The question measured F/V intake, “Did you eat five servings of vegetables and fruits per day (the total amount of vegetables and fruits should be 400 g, with 80 g per serving) in the past two months?” which was adapted from the Diabetes Risk Questionnaire for the Chinese population survey, which showed a good test-retest reliability (0.988) [40].

#### 2.2.5. WC

WC was measured by trained nurses along the horizontal plane halfway between the lowest rib and the iliac crest. According to the WC criterion for Asians, a WC ≥ 80 cm in women is classified as central obesity (central obesity) [41].

#### 2.2.6. BMI

BMI was calculated using the formula BMI = kg/m^2^. Weight and height were measured on a balance weight scale with a height meter, with a lever balance to the nearest 0.1 kg and 0.5 cm without shoes. The study participants wore lightweight clothes and took off their shoes during weight and height measurements. According to the BMI classifications for Chinese, BMI value ≥ 24 and <27.9 kg/m^2^ was classified as overweight, and a BMI ≥ 28 kg/m^2^ was considered obese [41].

### 2.3. Data Collection

The research assistants distributed questionnaires at the research sites, conducted physical examinations, and collected blood samples on-site after the participants completed the cancer screening program. The participants completed an online questionnaire by themselves through an application system (https://www.wjx.cn/vm/P70pxUu.aspx, accessed on 1 July 2018). The research assistant explained the questions and helped operate the questionnaire application system when needed.

### 2.4. Statistical Methods

Descriptive analyses were used to describe the sample characteristics. Frequencies and percentages were used for categorical variables, and means and standard deviations (*SD*s) were calculated for numerical variables. Binary logistic regression and multiple interaction models were used to explore the independent (menopausal status, F/V intake, and sedentary time) and joint associations (menopausal status × F/V intake and menopausal status × sedentary time), respectively, with central obesity. The three-category menopausal status and two-category sedentary time variables were used to determine the joint menopausal status and sedentary time, which were divided into six categories: pre-menopause/≤4 h, peri-menopause/≤4 h, post-menopause/≤4 h, pre-menopause/>4 h, peri-menopause/>4 h, and post-menopause/>4 h. The three-category menopausal status and two-category F/V intake variables were used to identify the joint menopausal status and F/V intake, which were divided into six categories: pre-menopause/no five servings of F/V intake per day, peri-menopause/no five servings of F/V intake per day, post-menopause/no five servings of F/V intake per day, pre-menopause/five servings of F/V intake per day, peri-menopause/five servings of F/V intake per day, post-menopause/five servings of F/V intake per day. 

Based on the literature, age, marital status, education, family income, occupation, and chronic disease were all controlled for in the model. The level of significance was set at *p* < 0.05 in all tests. Statistical analysis was performed using the SPSS package (Version 24.0, IBM Corp., Armonk, NY, USA) for Windows.

## 3. Results

### 3.1. Demographics of Participants

Of 400 Chinese women invited to the study, 387 agreed to participate (96.8%). The mean age of the women was 47.7 years (*SD* = 6.6). Most women were married (97.9%) and received 9 years of education or less (79.8%). Approximately 73.6% of the women had a relatively low household income of less than 3000 RMB/month. More than half of the women were farmers (66.5%). A total of 2.8% of the women reported smoking. One woman reported drinking daily, 16.5% seldom drank, and the others never drank. About 36.6% of the women reported having a chronic disease (such as hypertension, type 2 diabetes mellitus, hyperlipidemia, kidney disease, cancer, myocardial infarction, or stroke). About 33.8% of the women were overweight, and 7.0% were obese (Table 1). No significant difference was observed in the demographic parameters between women with and without central obesity except for their age and BMI. 

### 3.2. Menopausal Status, Sedentary Time, and F/V Intake

A total of 47.4% (*n* = 183), 17.4% (*n* = 67), and 35.2% (*n* = 137) of the women reported to be in pre-, peri-, and post-menopausal stages, respectively. In addition, 60.2% (*n* = 233) of women had >4 h of sedentary time per day, and 44.8% (*n* = 173) did not consume five servings of F/V per day. No significant difference was observed in the sedentary time between women with and without obesity. A significant difference in F/V intake was noticed in women with and without central obesity. A total of 48.1% (*n* = 103) of women with central obesity and 51.9% (*n* = 111) of those without central obesity consumed five servings of F/V per day (*p* < 0.05) (see Table 2).

### 3.3. The Proportion of Women with Central Obesity

The proportion of women with central obesity among the participants was 52.8% (*n* = 145). A significant difference in central obesity prevalence was found with menopausal status. A total of 63.2% (*n* = 87), 59.7% (*n* = 40), and 42.6% (*n* = 18) of post-, peri-, and pre-menopausal women had central obesity, respectively (*p* < 0.05).

### 3.4. Independent Association of Sedentary Time, F/V Intake, and Menopausal Status with Central Obesity among Women 

Menopausal status was significantly associated with central obesity across four models (Model 1–4, Table 3), with the OR of central obesity increasing to 1.759–2.377 in peri- and post-menopausal women compared with pre-menopausal women (*p* < 0.05). The OR of central obesity in women who did not consume five servings of F/V per day increased to 1.527–1.607 compared with that of women who consumed five servings of F/V per day (*p* < 0.05). 

In the unadjusted and multivariate-adjusted models, no significant independent association was found between sedentary time and central obesity (Models 1–4; Table 3). 

### 3.5. Joint Association of Sedentary Time or F/V Intake and Menopausal Status among Women with Central Obesity 

Compared with pre-menopausal women who did not have five servings of F/V intake per day, whether confounding factors were controlled for or not, pre-menopausal women who had five servings of F/V intake per day had a significant association with a low risk of central obesity in the three models (mutual OR: 0.444; 95% CI: 0.236–0.837; *p* < 0.05). However, no difference was found in peri- and post-menopausal women (*p* > 0.05).

Compared with pre-menopausal women who had >4 h of sedentary time per day, peri- and post-menopausal women who had >4 h of sedentary time per day had a significant association with a high risk of central obesity in the unadjusted (OR: 2.615; 95% confidence interval (CI): 1.084–6.307 in peri-menopause and OR: 2.996; 95% CI: 1.498–5.994 in post-menopause; *p* < 0.05) and multivariate-adjusted models (mutual OR: 2.466; 95% CI: 0.984–6.182 in peri-menopause and mutual OR: 2.274; 95% CI: 1.046–4.943 in post-menopause; *p* < 0.05), whereas no significant association was found in pre-menopausal women who had >4 h of sedentary time per day (OR: 1.202–1.508; *p* > 0.05) (see Table 4). 

## 4. Discussion

To the authors’ knowledge, this research was the first to explore the associations between sedentary time or F/V intake and central obesity among women in different menopausal statuses. Menopausal status was a significant independent factor associated with central obesity. Among peri- and post-menopausal women, >4 h of sedentary time was a risk factor for developing central obesity, whereas, among pre-menopausal women, the consumption five servings of F/V per day was the protective factor against central obesity. Based on our findings, targeted prevention strategies of central obesity could be recommended for women of different menopausal statuses. We also had several significant results. First, we found that >4 h of sedentary time was significantly associated with a higher risk of central obesity among peri- and post-menopause women. Taking five servings of F/V per day was significantly associated with a lower risk of central obesity among pre-menopause women. Second, we found that over half of women had central obesity; most women had >4 h of sedentary time and didn’t take five servings of F/V per day. Third, peri-menopause, post-menopause, and not taking five servings of F/V per day were independent risk factors of central obesity. 

The high prevalence of central obesity in Chinese women, especially peri- and post-menopausal women, is alarming. This study’s results were similar to the 2011 China Health and Nutrition Survey findings, which indicated that 51.7% of women aged 20 years or older had central obesity [5]. However, the prevalence of central obesity in women in Jiangxi Province, China was lower at 29.0% [42], which can be interpreted in light of the lower economic development level of Jiangxi province (per capita gross domestic product is $5100 versus the average of China in that year ($7485) [42,43]). People who live in developed cities may be more vulnerable to globalization, leading to changes in eating patterns characterized by low consumption of fruits and green leafy vegetables and high consumption of animal proteins and calories [44]. Meanwhile, people living in provinces with a lower economic status generally have low-paying jobs that typically involve physically demanding work [44]. In this study, about 80% of women received nine years of education or less in this study which is consistent with the average level in China, as people are required to attend nine years of schooling in China [45]. Interventions need to consider the social determinants of health factors and be tailored to the culture, health literacy level, and needs of Chinese women to reduce health issues related to central obesity.

This study revealed that peri- and post-menopausal women in China were likely to have central obesity. This result was consistent with a study in Southern Brazil, which reported that post-menopausal women have five times more likely to have central obesity than pre-menopausal ones [46]. Nevertheless, menopausal status has been associated with increased abdominal subcutaneous and visceral fat, possibly due to the absence of estrogens [47]. 

Fewer women with central obesity had five servings of F/V intake per day compared with those without central obesity in our study, indicating that five servings of F/V intake may benefit central obesity prevention [48]. This finding was consistent with a prospective study in Korea [49]. However, the five servings of F/V intake per day was significantly associated with a low risk of central obesity only among pre-menopausal women in this study. F/V may protect against central obesity or weight gain by reducing the energy density of the diet, reducing total fat intake [34], or through abundant soluble dietary fibers, which could lower energy absorption [50]. F/V is also rich in phytochemicals that have antioxidative and anti-inflammatory effects against obesity-induced oxidative stress and subclinical inflammation [51]. It is possible that the adipocytes of pre-menopausal women are more sensitive to the phytochemicals in F/V than those of peri- and post-menopausal women. Future studies need to include more specific dietary information to determine the sources of calorie intake and sensitivity of F/V intakes in metabolic rate and obesity risk among women in different menopause status. 

This study revealed that >4 h of sedentary time per day had no significant independent association with central obesity among women in the pre-menopause stage. This finding was inconsistent with the previous review indicating that a long sedentary time was a risk factor for central obesity in women [35]. However, our study revealed that the association between sedentary time and central obesity was influenced by menopausal status. Sedentary time > 4 h per day was associated with a high risk of central obesity among peri- and post-menopausal women. Compared with peri- and post-menopausal women, pre-menopausal women are usually younger and have a higher level of estrogens [1,52]. Peri- and post-menopausal women may spend more time in sedentary behaviors than pre-menopausal women; older women are more likely to spend more time in sedentary behaviors because of their higher retirement rate than younger women [53]. On the other hand, estrogen depletion also leads to inactivity [54] and is related to prolonged sedentary time [55], which can also explain why peri- and post-menopausal women may spend more time in sedentary behavior. Moreover, an animal study indicated that exercise training strongly influences lowering body fat accumulation following a decrease in estrogen levels [56]. Thus, peri- and post-menopausal women may accumulate more body fat with equal sedentary time compared with pre-menopausal women. 

### 4.1. Study Limitations

This study has several limitations. First, this research utilized a cross-sectional design, and thus, causal relationships between sedentary time or F/V intake and central obesity cannot be fully established. Second, we did not identify the menopausal status using a biological matrix (such as estradiol and follicle-stimulating hormone level) nor examine the effect of contraceptives and menopausal treatment, which may also influence weight gain. Third, we used self-reported sedentary time and F/V intake. Thus, this research may have recall and reporting bias. Fourth, given the small number of women in peri-menopausal status who had >4 h of sedentary time, the conclusions should be considered carefully with caution. Fifth, we did not include women younger than 35 years or older than 64 years and women who did not attend the universal screening, in addition to different percentages of women in pre-, peri-, and post-menopausal stages in this study. Thus, the findings of this study may not be generalized to other populations. Lastly, the survey only assessed the last two months of sedentary behavior and F/V intake, which may not accurately measure the long-term lifestyle of the participants.

### 4.2. Implication

In terms of clinical implication, given that central obesity is prevalent among women in the central-south region of China, interventions to reduce the incidence of this condition are warranted. Interventions targeting peri- and post-menopausal women and reducing excessive sedentary time to ≤4 h are also needed. Studies with a biological matrix that can be used to identify the menopausal status can be considered in the future. Since pre-menopausal women are sensitive to F/V intake, they should be encouraged to consume five servings of F/V per day. In terms of research implications, randomized controlled clinical trials are needed to rigorously assess the causal relationship of these behaviors with central obesity among Chinese women. 

## 5. Conclusions

Central obesity is common in the central-south region of China. Most women spend > 4 h in sedentary behavior and do not consume the recommended F/V daily intake, identified as risk factors for central obesity. For peri- and post-menopausal women, interventions aiming to reduce sedentary time to ≤4 h are especially recommended to decrease the risk of central obesity. In addition, healthier lifestyle intervention is warranted to prevent central obesity development, including reducing the sedentary time to <4 h per day for peri- and post-menopausal women, while taking five servings of F/V per day for pre-menopausal women.

## Figures and Tables

**Table 1 ijerph-19-10083-t001:** Demographic Characteristics of the Study Sample.

Characteristic	Number of Samples (*n*)	%
Age		
≤44	124	32.0
45–54	205	53.0
≥55	58	15.0
Marital Status		
Married	379	97.8
Unmarried	2	0.6
Divorced	3	0.8
Widowed	3	0.8
Education		
≤9 years	308	79.8
>9 years	78	20.2
Family Income		
≤3000 RMB	220	57.0
>3000 RMB	167	43.0
Occupation		
Farmer	258	66.7
Factory worker	44	11.4
Office worker	62	16.0
Other	23	5.9
Family members		
Living alone	2	0.5
Core family	247	64.2
Living with parents	138	35.3
Drinking		
Often	1	0.3
Seldom	64	16.5
Never	322	83.2
Smoking		
Yes	11	2.8
No	376	97.2
Family member smoking		
Yes	275	71.9
No	108	28.1
Missing	4	
Medical history (Chronic illness)		
Yes	142	36.6
No	246	63.4
BMI		
≤23.9	228	59.2
24.0–27.9	130	33.8
≥28.0	27	7.0

**Table 2 ijerph-19-10083-t002:** Description of menopause status, F/V intake, and sedentary time of women with and without central obesity.

Characteristic	Number of Samples (n)	%	WC		*p* Value
			% ≤79 cm	% ≥80 cm	
Menopausal status					
Pre-menopause	183	47.4	57.4	42.6	*p* = 0.00
Peri-menopause	67	17.4	40.3	59.7	
Post-menopause	137	35.2	36.8	63.2	
Have five servings of F/V intake/day					
Yes	214	55.2	51.9	48.1	*p* = 0.04
No	173	44.8	41.4	58.6	
Sedentary time more than 4 h/day					
Yes	233	60.2	49.8	50.2	*p* = 0.25
No	154	39.8	43.8	56.2	

**Table 3 ijerph-19-10083-t003:** Adjusted OR for the independent association of menopausal status, F/V intake and sedentary time of women with central obesity.

	N (%)	Model 1 OR (95% CI)	Model 2 OR (95% CI)	Model 3 OR (95% CI)	Model 4 OR (95% CI)
Menopausal status					
Pre-menopause	184 (47.5)	1	1	1	1
Peri-menopause	67 (17.3)	2.013 * (1.140–3.577)	2.093 * (1.141–3.841)	2.107 * (1.146–3.872)	2.045 * (1.109–3.376)
Post-menopause	136 (35.2)	2.337 * (1.483–3.684)	1.804 * (1.027–3.168)	1.824 * (1.037–3.208)	1.780 * (1.009–3.140)
Five servings of F/V intake/day					
Yes	214 (55.3)	1	1	1	1
No	173 (44.7)	1.527 * (1.020–2.286)	1.593 * (1.041–2.437)	1.545 * (1.005–2.376)	1.540 * (1.001–2.370)
Sedentary time					
≤4 h	233 (60.2)	1	1	1	1
>4 h	154 (39.8	1.251 (0.831–1.884)	1.321 (0.860–2.029)	1.340 (0.869–2.068)	1.334 (0.863–2.064)

*: *p* < 0.05; Model 1: unadjusted for any variable, Model 2: adjusted for age, marital status, education, family income, occupation, and medical history. When the effect of menopausal status on central obesity was analyzed, Model 3, based on Model 2, was additionally adjusted for sedentary time, and Model 4 was additionally adjusted for F/V intake/day. When the effects of five servings of F/V intake/day and sedentary time on central obesity were analyzed, Model 3 was additionally adjusted for menopausal status based on Model 2, whereas F/V intake/day and sedentary time were controlled based on Model 3 for Model 4, respectively.

**Table 4 ijerph-19-10083-t004:** Joint association of F/V intake or sedentary time and menopausal status with central obesity in women.

	N (%)	Model 1 OR (95% CI)	Model 2 OR (95% CI)	Model 3 OR (95% CI)
Menopausal status and Five servings of F/V intake/day				
Pre-menopause, No Five servings of F/V intake/day	78 (20.2)	1	1	1
Peri-menopause, No Five servings of F/V intake/day	34 (8.8)	1.458 (0.641–3.317)	1.449 (0.612–3.429)	1.427 (0.602–3.385)
Post-menopause, No Five servings of F/V intake/day	61 (15.8)	1.600 (0.806–3.177)	1.162 (0.531–2.543)	1.169 (0.533–2.562)
Pre-menopause, Five servings of F/V intake/day	106 (27.4)	0.484 * (0.266–0.880)	0.461 * (0.246–0.863)	0.456 * (0.243–0.856)
Peri-menopause, Five servings of F/V intake/day	33 (8.5)	1.225 (0.539–2.784)	1.241 (0.522–2.955)	1.265 (0.531–3.016)
Post-menopause, Five servings of F/V intake/day	75 (19.4)	1.515 (0.795–2.888)	1.115 (0.525–2.370)	1.122 (0.527–2.387)
Menopausal status and Sedentary time				
Pre-menopause, ≤4 h	108 (27.9)	1	1	1
Peri-menopause, ≤4 h	41 (10.6)	2.307 * (1.109–4.801)	2.341 * (1.085–5.048)	2.335 * (1.078–5.056)
Post-menopause, ≤4 h	86 (22.2)	2.758 * (1.536–4.950)	2.103 * (1.079–4.098)	2.068 * (1.058–4.043)
Pre-menopause, >4 h	75 (19.4)	1.508 (0.830–2.740)	1.532 (0.827–2.838)	1.542 (0.830–2.864)
Peri-menopause, >4 h	26 (6.7)	2.615 * (1.084–6.307)	2.843 * (1.136–7.112)	2.672 (1.061–6.729)
Post-menopause, >4 h	51 (13.2)	2.996 * (1.498–5.994)	2.422 * (1.113–5.270)	2.401 * (1.098–5.250)

*: *p* < 0.05; Model 1: unadjusted for any variable, Model 2: adjusted for age, marital status, education, family income, occupation, and medical history; Model 3: based on Model 2, F/V intake/day and sedentary time were controlled, respectively.

## Data Availability

The datasets generated during and/or analyzed during the current study are available from the corresponding author on reasonable request.

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
