# Peer review of "Does the Association of Sedentary Time or Fruit/Vegetable Intake with Central Obesity Depend on Menopausal Status among Women?"

_ijerph, 2022, doi:10.3390/ijerph191610083_

Round 1
Reviewer 1 Report
Dear Editor,
Thank you very much for your trust and for sending the manuscript entitled The work concerns a very interesting topic of the influence of fruit, vegetable and sedentary consumption on obesity in postmenopausal women. Introduction The introduction introduces the reader to the most important information. The purpose was clearly presented. In the material and methodology, the authors discuss the inclusion and exclusion criteria from the study, present the consent of the local bioethical committee, and the calculations of the minimum sample size. Jasmo describes the statistical analyzes used. Results In the results, some doubts are raised by the education of the respondents. Such a high compliance of education may cause disturbances in conclusions about the entire population. Please explain that. The very large variety of women included in the study at different stages of menopause may affect the results obtained. Tables with the results described in the text should be placed directly below the description, which makes it easier to read and interpret the data. The correlation between the onset of menopause and obesity has been known for many years. Interesting discussion, but the authors refer to a rather old bibliography.
Author Response
Thanks for your comments, attached is our reply.

Reviewer 2 Report
Summary
Given the rapid increase in central obesity among women, it is important to identify risk factors and possible intervention strategies. It is still debated how menopausal status modifies the known risk factors sedentary time and fruit/vegetable intake. In the present manuscript, Su and colleagues aim to investigate the effect of menopausal status on the association of sedentary time or fruit/vegetable intake with central obesity. They performed a cross-sectional study in Hunan, China, using a self-report questionnaire to assess demographic characteristics and the factors of interest, i.e. menopausal status, sedentary time and fruit/vegetable intake. Among the 387 women included in the study, more than 50% presented with central obesity, had an increased sedentary time, and did not consume the recommended 5 servings of fruit/vegetables per day. Peri- and postmenopausal status as well as reduced fruit/vegetable intake were identified as risk factors for central obesity while sedentary time was not significantly associated with central obesity. Of note, among peri- and postmenopausal women, increased sedentary time was linked to higher risk of central obesity and in pre-menopausal woman, the consumption of 5 portions of fruits/vegetables per day was associated with a low risk of central obesity. The authors conclude that reducing sedentary time is a promising strategy to prevent central obesity in peri- and postmenopausal women whereas for pre-menopausal women they recommend an increase in fruit/vegetable intake.
Comments
Overall, the manuscript is well written and the rationale for the study is layed out clearly. Its findings confirm the results of previous studies, but do not allow to make detailed conclusions about the role of menopausal status in central obesity in association with sedentary time and fruit/vegetable intake. There are some points of concern that should be addressed to improve the manuscript:
Broad comments:
1) The reference numbers in the text do not match the numbers in the reference section, therefore it is hard to evaluate whether the cited literature is appropriate. This must be corrected.
2) Supplementary data are not related to the study.
3) Minor spell and grammar checks are needed.
Specific comments:
1) Introduction:
· line 57-58: It would be beneficial to state some reference percent values: how much was the increase in the percentage of women with central obesity?
· line 114: remove “among pre-menopausal women”
2) Methods:
· Women older than 60 years were excluded from the study. Is there a reason for the excluding this age group?
· Although the medical history was assessed in the survey, it was not included in the analysis. Could the authors explain why they did not further report on health status and medication since these might also be important confounders.
3) Results:
· line 232: The authors state that 33.8% of the participating women were overweight and 7% were obese. These numbers are not included in Table 1 as indicated. Please include the respective BMI values in the Table.
· The way the results are presented in 3.4 (line 251 and following) are confusing for the reader and it would be beneficial to edit this paragraph.
4) Discussion:
· In the first paragraph of the discussion, the authors state that “reducing the sedentary time to <4h was important to fight against central obesity”. However, there is no evidence in this study to make this statement, since the effect of reducing sedentary time was not tested here. Therefore, this statement should be re-phrased.
· In lines 324-326, sensitivity of adipocytes to fruit/vegetable intake is mentioned. How are adipocytes related to fruit/vegetable intake? Is there anything known in the literature?
· Limitations: Next to the limitations already mentioned in the discussion section, there are two other main limitations that need to be mentioned: i) the study does not control for health status and medication, including f.ex. contraceptives, ii) the survey only assesses the last two months of sedentary behavior and fruit/vegetable intake.
5) Conclusions:
The authors conclude that “the consumption of >5 servings of fruit/vegetables per day should be prioritized to avoid the development of central obesity among pre-menopausal women”. However, with the present data, it is not clear whether fruit/vegetable consumption alone are sufficient to prevent central obesity. More likely, a healthier lifestyle in general rather than exclusively increasing the fruit/vegetable intake will be more efficient. Therefore, this conclusion should be made with caution.
Author Response

(The authors gave the same response as above.)

Reviewer 3 Report
The title as well as the introduction raised expectations about your manuscript and research. The topic you are addressing would be a relevant addition to existing literature. Thank you for this valuable contribution. I will structure my feedback in (a) general remarks (these comments cover feedback applicable in the entire manuscript), and (b) specific remarks (feedback on sentence and/or word level). The specific remarks can include a quote from your original manuscript to refer to a specific section. The specific remarks will refer to page (emphasis added in boldface; e.g., 1.15/16) and row(s; e.g., 11.15/16).
General remarks:
The overall manuscript is neat and written concisely—with relevant information for existing literature. The study involves concepts relevant for practice. My compliments. I am wondering if you can write the abstract more concisely. It is rather long.
Specific remarks:
p.1.17/18 With your current sentence construction it is unclear that “differs by menopausal status” refers to the prevalence of central obesity.
p.1.32 There is a tab (or at least extra space) after “17.4%”.
p.2.58–64 Avoid back-to-back brackets: (x)[y].
p.2/3 Information presented is clear and formally written.
p.3.133 I am wondering if the universal cancer screening programs attracts women with specific characteristics (i.e., can a bias be present here?). I assume that this program also conducts analyses to check what women—with subsequent characteristics—visit the screening.
p.4.143 There is extra space before “Women”.
p.4.152 This exclusion criterium can contain more examples. What do you consider limited physical activity? You can link this to x minutes per day.
p.4.154 In a similar vein, what do you consider a restricted diet? I assume you know what you consider that, but at this point I do not have that information. For example, if the individual cannot eat gluten do you consider that restricted? What about lactose? Nuts? Soy? These exclusion criteria need to be explained. You can add an appendix with examples.
p.4.157 The “N”: is that the total amount of participants? In addition, this has to be presented in italics.
p.4.161 You need to insert a space between “2.2” and “Measurement”.
p.5.209 There is a space (or tab) before “, respectively”.
p.5.227 SD needs to be presented in italics.
p.6.236 The “n” needs to be placed in italics. Also avoid the back-to-back brackets in that paragraph.
p.Table1 The tables are difficult to read due to the spacing. Can you make it smaller so it fits on one page? In addition, this information would be suitable in an appendix
p.Table2 In a similar vein, the table is unreadable. Shorten the table so they can be presented on one (or half a) page.
p.12.316 Extra space.
p.all You present the “>4” sometimes with and sometimes without spaces. Keep this consistent.
References The hyphen between the page numbers should be replaced by an en dash. Moreover, the capital letter use in the titles is inconsistent (sometimes you capitalize each essential word (see reference 4) and sometimes you do not (see reference 2). Make this consistent, please. Furthermore, your display of page numbers is inconsistent: p. 337-43 (reference 32) versus p. 251-258 (reference 36). Use one wat of displaying this and apply this consistently.
Author Response

(The authors gave the same response as above.)
